# VRM: Knowledge Distillation via Virtual Relation Matching

## Abstract

Knowledge distillation (KD) aims to transfer the knowledge of a more capable yet cumbersome teacher model to a lightweight student model. In recent years, relation-based KD methods have fallen behind, as their instance-matching counterparts dominate in performance. In this paper, we revive relational KD by identifying and tackling several key issues in relation-based methods, including their susceptibility to overfitting and spurious responses. Specifically, we transfer novelly constructed affinity graphs that compactly encapsulate a wealth of beneficial inter-sample, inter-class, and inter-view correlations by exploiting virtual views and relations as a new kind of knowledge. As a result, the student has access to rich guidance signals and stronger regularisation throughout the distillation process. To further mitigate the adverse impact of spurious responses, we prune the affinity graphs by dynamically detaching redundant and unreliable edges. Extensive experiments on CIFAR-100, ImageNet, and MS-COCO datasets demonstrate the superior performance of the proposed virtual relation matching (VRM) method over a range of tasks, architectures, and set-ups. For instance, VRM for the first time hits 74.0% accuracy for ResNet50→MobileNetV2 distillation on ImageNet, and improves DeiT-Ti by 14.11% on CIFAR-100 with a ResNet56 teacher. Thorough analyses are also conducted to gauge the soundness, properties, and complexity of our designs. Code and models will be released.

## 1 Introduction

Deep learning is achieving incredible performance at the cost of increasing model complexity and overheads. As a consequence, large and cumbersome neural models struggle to work in resource-constrained environments. Knowledge distillation (KD) has been proposed by Hinton et al. (2015) to address this issue by transferring the knowledge of larger and more capable models to smaller and lightweight ones that are resource-friendly. KD work by minimising the distance between compact representations of knowledge extracted from the teacher and student models. According to the type of such knowledge representations, KD methods can be broadly categorised into feature-based (Romero et al., 2015), logit-based (Hinton et al., 2015), and relation-based (Park et al., 2019) approaches. The former two directly match the feature maps or logit vectors produced by the teacher and student models for each training sample, which is essentially instance matching (IM). By contrast, relation matching (RM) methods construct and match structured relations extracted within a batch of model responses. A conceptual illustration is presented in Figs. 1a and 1b.

Instance matching has been the prevailing distillation approach in recent years. Popular KD benchmarks see a dominance by IM-based methods such as FCFD (Liu et al., 2023a), NORM (Liu et al., 2023b), and TGeoKD (Hu et al., 2024), with many different downstream tasks successfully tackled by directly adopting IM-based distillation (Wang et al., 2019; Yang et al., 2023a; Chang et al., 2023; Chen et al., 2023). Yet, recent studies discovered that relational knowledge is more robust to variations in neural architectures, data modalities, and tasks (Park et al., 2019; Tung & Mori, 2019). Meanwhile, methods transferring relations have also achieved promising performance for a range of tasks, including but not limited to segmentation (Yang et al., 2022a) and detection (Chong et al., 2022; Jang et al., 2024).

Despite growing interest, relation-based methods still fall significantly short compared to their instance matching counterparts. Even the strongest RM method has been outperformed easily by

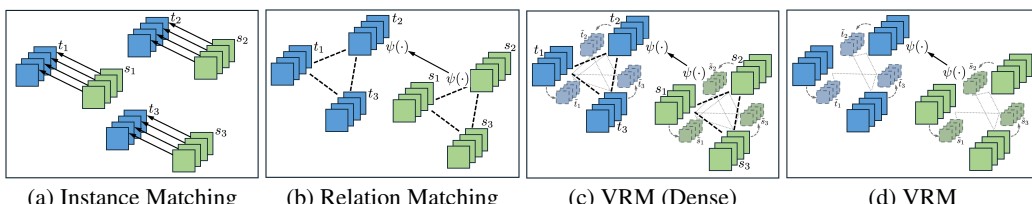

| (a) Instance Matching | (b) Relation Matching | (c) VRM (Dense) | (d) VRM |

Figure 1: Conceptual illustration of the proposed VRM compared to existing KD methods based on instance matching and relation matching.

recent IM solutions (Huang et al., 2022) (see Tabs. 1 and 2). RM-based methods also struggle with more challenging tasks such as object detection (Huang et al., 2022). Moreover, previous RM-based methods are primarily limited to matching inter-sample (Tung & Mori, 2019; Passalis & Tefas, 2018; Huang et al., 2022), inter-class (Huang et al., 2022), or inter-channel (Yim et al., 2017; Liu et al., 2021a) relations via simple Gram matrices. To our best knowledge, no different forms of relations other than the above have been proposed since 2022.

This paper fills this gap with a new kind of relations for KD – *inter-view* relations (Fig. 1c), which seamlessly and compactly integrate with previous inter-sample and inter-class relations. Our designs are motivated by two important observations made about RM methods in a set of pilot experiments: 1) RM methods are more susceptible to overfitting than IM methods; 2) RM methods are subject to an adverse gradient propagation effect. We empirically find that incorporating richer and more diverse relations into the matching objective helps mitigate both issues. To this end, we generate *virtual views* of samples through simple transformations, followed by constructing *virtual affinity graphs* and transferring the *virtual relations* between real and virtual samples along the edges. In lieu of Gram matrices that suffer from significant knowledge loss (Tung & Mori, 2019; Passalis & Tefas, 2018; Huang et al., 2022), we preserve the raw relations along the secondary dimension as auxiliary knowledge which adds to the types and density of relational knowledge transferred. Moreover, we also prune our affinity graphs by striping away both redundant and unreliable edges to further alleviate the propagating gradients of spurious samples (Fig. 1d).

The above insights and remedies altogether lead to a novel *Virtual Relation Matching* (VRM) framework for KD. VRM is conceptually simple, easy to implement, and devoid of complicated training procedures. It is capable of transferring rich, sophisticated knowledge robust to overfitting and spurious signals. VRM sets new state-of-the-art performance on CIFAR-100 and ImageNet datasets for different ConvNet and Transformer architectures. On MS-COCO object detection, VRM for the first time performs competitively to high-performance IM solutions by distilling purely relational knowledge. Perhaps more significant is that VRM makes relation-based methods regain competitiveness and back in the lead over instance matching approaches across various tasks and settings. We will release the code and models to encourage further endeavours in relation-based KD.

To summarise, the contributions of this work include:

- We make an early effort to present comparative analyses of existing KD methods through the lens of training dynamics and sample-wise gradients, and identify overfitting and spurious gradient diffusion as two main cruxes in relational KD methods.
- We distill richer and more diverse relations by generating virtual views, constructing virtual affinity graphs, and matching virtual relations. We also for the first time approach relational KD with considerations of spurious samples and gradients by pruning redundant and unreliable edges and other designs to relax the matching criterion.
- We present the streamlined VRM framework for knowledge distillation, with extensive experimental results on a diversity of neural architectures and tasks to highlight its superior performance, alongside rigorous analyses on the soundness and efficiency of our designs.

## 2 RELATED WORK

**KD via instance-wise transfer.** Knowledge distillation (KD) was first proposed in Hinton et al. (2015), where the student is trained to match its predictions to those of the teacher for each sam-

ple. Follow-up works have mostly followed such *instance matching* (IM) paradigm, and can be categorized into logit (or prediction)-based and feature-based methods according to what is matched. Since Hinton et al. (2015), logit-based KD has evolved from using adaptively-softened logits (Li et al., 2023b) or an ensemble of differently-softened (Jin et al., 2023) distributions to decoupling target- and non-target logits (Zhao et al., 2022; Yang et al., 2023b) and applying logit transformation (Sun et al., 2024; Zheng & Yang, 2024). Methods such as TAKD (Mirzadeh et al., 2020) and DGKD (Son et al., 2021) set up auxiliary ad-hoc networks between teacher and student to facilitate logit transfer. Early feature-based methods (Romero et al., 2015; Ahn et al., 2019; Heo et al., 2019a) directly minimise the distance between the feature maps at a specified layer in both the teacher and student networks. AT (Zagoruyko & Komodakis, 2017) and CAT-KD (Guo et al., 2023) transfer the salient regions in features. Sophisticated distillation paths have been designed, such as "many-to-one" layer-wise matching in SemCKD (Chen et al., 2021a) and "one-to-many" patch matching in TaT (Lin et al., 2022). All these methods are based on instance-wise transfer of knowledge, either logits or features, and are herein referred to as the "IM" methods, as illustrated by Fig. 1a.

**KD via relation transfer.** Several works attempt to transfer instead mutual relations or correlations mined amongst the network outputs extracted from a batch of training instances. These *relation matching* (RM) methods usually involve constructing network outputs into compact relation representations that encodes rich higher-order information, as depicted in Fig. 1b. Different relation encoding functions may be used, including inter-sample (Park et al., 2019; Passalis & Tefas, 2018; Peng et al., 2019; Tung & Mori, 2019; Huang et al., 2022), inter-class (Huang et al., 2022), inter-channel (Liu et al., 2021a), and inter-layer (Yim et al., 2017), and contrastive (Tian et al., 2020) relations. To date, IM solutions have dominated KD with their superior performance, leading top-performing relation-based methods by a considerable margin. While many new IM methods are found within the last two years, frustratingly few new RM solutions are being proposed. In this work, we strive to close this gap with a new kind of relations for KD — inter-view virtual relations, and revive relation-based KD by making it overtake its IM counterparts.

**Learning with virtual knowledge.** While not a standalone research topic, learning with virtual knowledge finds relevance in a variety of learning-based problems. For instance, a commonly used paradigm in 3D vision tasks is to learn (or construct) from the raw data a virtual view or representation as auxiliary knowledge in solving the main task, which range from object reconstruction (Carreira et al., 2015) and optical flow (Aleotti et al., 2021) to 3D semantic segmentation (Kundu et al., 2020), monocular 3D object detection (Chen et al., 2022a), and 3D GAN inversion (Xie et al., 2023). More broadly, many data-efficient learning methods also share the spirit of utilising virtual knowledge. For instance, self-supervised learning methods generate virtual views of the unlabelled data to enable the learning of pretext tasks (Gidaris et al., 2018; Chen et al., 2020b). Another popular paradigm is transformation-invariant representation learning (Misra & Maaten, 2020; Sohn et al., 2020) in semi-supervised learning and domain adaptation. It enforces consistency between representations learnt for a raw sample and a virtual view of it. The virtual view is often obtained by applying semantic-preserving transformations to the raw sample (Cubuk et al., 2020; 2019). This work is more related to this later paradigm, but involves learning with virtual knowledge in a different context, via different approaches, and for a different problem.

## 3 METHOD

### 3.1 PRELIMINARIES

KD methods generally employ a cross-entropy (CE) loss and a distillation loss to supervise the student learning. The CE loss is computed between the student logits $\mathbf{z}_i^s$ for each sample and its ground-truth label $\mathbf{y}_i$. The distillation loss matches teacher and student outputs via a distance metric $\phi(\cdot)$. For instance, in vanilla KD (Hinton et al., 2015) $\phi(\cdot)$ is the Kullback–Leibler divergence (KLD) between teacher logits $\mathbf{z}^t$ and student logits $\mathbf{z}^s$:

$$\mathcal{L}_i^{\text{KD}} = \phi_{\text{KLD}}(\mathbf{z}_i^s, \mathbf{z}_i^t) = \tau^2 \sum_{j=1}^{C} \sigma_j(\mathbf{z}_i^t/\tau) \log \frac{\sigma_j(\mathbf{z}_i^t/\tau)}{\sigma_j(\mathbf{z}_i^s/\tau)}, \tag{1}$$

where $\sigma(\cdot)$ is the Softmax operation with temperature parameter $\tau$, and $C$ is the number of classes. For feature-based methods, $\phi(\cdot)$ can be the mean squared error (MSE) between teacher and student feature maps for each instance (Romero et al., 2015), *i.e.*, $\mathcal{L}_i^{\text{KD}} = \phi_{\text{MSE}}(\mathbf{f}_i^s, \mathbf{f}_i^t)$.

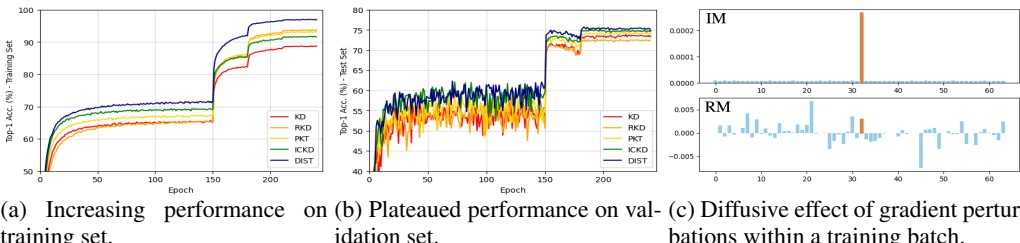

(a) Increasing performance on training set.

(b) Plateaued performance on validation set.

(c) Diffusive effect of gradient perturbations within a training batch.

Figure 2: Conceptual illustration of the proposed method compared to existing KD methods based on instance matching and relation matching.

For relation-based methods, a relation encoding function $\psi(\cdot)$ first abstracts teacher or student outputs of all instances within a training batch into a relational representation, before applying $\phi(\cdot)$ to match these relational representations between the teacher and the student. As an example, the KD objectives of DIST (Huang et al., 2022) take the general form of:

$$\mathcal{L}^{\text{KD}} = \phi(\psi(\mathbf{z}_1^s, \mathbf{z}_2^s, ..., \mathbf{z}_B^s), \psi(\mathbf{z}_1^t, \mathbf{z}_2^t, ..., \mathbf{z}_B^t)), \tag{2}$$

where $B$ is the batch size, and subscript $i$ of $\mathcal{L}^{\text{KD}}$ is dropped because the loss is computed for a batch of instances. DIST employs both inter-class and inter-sample relation encoders as $\psi(\cdot)$.

## 3.2 PILOT STUDIES

**Training dynamics of KD methods.** We examine the training dynamics of different relation-based methods on CIFAR-100 using ResNet32×4 → ResNet8×4 as the teacher-student pair. In Figs. 2a and 2b, we immediately notice that relational methods achieve significantly higher training accuracy, but they only marginally lead or even fall short in test accuracy compared to IM-based KD (Hinton et al., 2015). We hypothesise that relational methods are more prone to overfitting. This is expected given that the optimality of IM matching (cond. $\mathcal{A}$) implies the optimality of relation matching (cond. $\mathcal{B}$), while the converse does not hold. Concretely, $\mathcal{A} \Rightarrow \mathcal{B} \ \wedge \ \neg \mathcal{B} \Rightarrow \neg \mathcal{A}$. In other words, relation matching is a weaker and less constrained objective than IM, which makes the student more readily fit the teaching signals and not generalise well. Thus, we conclude that *C1: relation matching methods are more prone to overfitting.*

**Gradient analysis of KD methods.** We investigate the gradient patterns within a batch when a spurious sample produces a major misguiding signal. To this end, we first generate two random vectors $\mathbf{x}, \mathbf{y} \sim \mathcal{N}(0, 1)$ for $\mathbf{x}, \mathbf{y} \in \mathbb{R}^{B \times D}$, where $B$ is the batch size and $D$ is the dimension of per-sample predictions, $\mathbf{x}$ is taken as the sample-wise predictions, and $\mathbf{y}$ the supervision signals. We then add a noise vector $\epsilon = c \cdot \mathbf{z}$ to $\mathbf{x}_t$, where $\mathbf{z} \sim \mathcal{N}(0, 1)$ and $c$ is a scaling factor. In this case, $\mathbf{x}_t + \epsilon$ becomes a spurious prediction within our batch. We compute the loss from $\mathbf{x}$ and $\mathbf{y}$ using either IM or RM objectives, and consider the change in sample-wise gradients $\mathbf{g}$ within the batch upon the injection of the spurious sample. Formally, we visualise

$$\Delta \mathbf{g} = \left[ \| \frac{\partial \mathcal{L}}{\partial(\mathbf{x}'_i)} \|_2 - \| \frac{\partial \mathcal{L}}{\partial(\mathbf{x}_i)} \|_2 \right]_{i=1}^B, \quad \text{s. t. } \mathbf{x}'_i = \mathbf{x}_i + \epsilon \cdot \mathbb{I}(i=t), \quad \mathcal{L} \in \{\mathcal{L}_{IM}, \mathcal{L}_{RM}\}.$$

In Fig. 2c ($B = 64$ and $t = 32$), when the IM objective is used, only the spurious sample receives a prominent gradient. Whereas for an RM objective, many other samples receive significant gradients as they are directly connected to $\mathbf{x}_t$ in the computational graph of the RM loss. In other words, the spurious signals produced by one malign prediction will propagate to and affect all samples within a batch (in fact, those closer to $\mathbf{x}_t$ within the prediction manifold are more strongly affected). This means other sample-wise predictions will be significantly updated only to accommodate a malign prediction, even if they are already in relatively good shape. Through this investigation, we discover that *C2: relation matching methods are more prone to the adverse impact of spurious samples.*

For *C1*, common approaches to combat overfitting include the incorporation of richer learning signals and regularisation, which for RM-based KD methods means richer relations constructed and transferred. For *C2*, an intuitive solution is to identify and suppress the effect of spurious predictions or relations or to slacken the matching criterion. Guided by these principles, let us move on to formally build up our method.

### 3.3 CONSTRUCTING INTER-SAMPLE RELATIONS

The remaining parts of this section present step-by-step descriptions of individual design choices that in tandem constitute our method. We first construct relation graph $\mathcal{G}^{IS}$ that encodes inter-sample affinity within a batch of sample predictions $\{\mathbf{z}_i\}_{i=1}^B$. Different from Park et al. (2019); Tung & Mori (2019), our relations are constructed from predicted logits which embed more compact categorical knowledge. We use the pairwise distance between instance-wise predictions within a batch as our measure of affinity. Existing methods leverage the Gram matrices (Peng et al., 2019; Passalis & Tefas, 2018; Tung & Mori, 2019; Huang et al., 2022) to encode inter-sample relations, but we find that this leads to collapsed inter-class knowledge via the inner product operation. Instead, our pairwise distance preserves the inter-class knowledge along the secondary dimension (*i.e.*, the class dimension), which enables such information to be explicitly transferred as auxiliary knowledge along with the matching of inter-sample relations.

Thus, we have constructed a dense relation graph $\mathcal{G}^{IS}$, which comprises $B$ vertices and $B \times B$ edges: $\mathcal{G}^{IS} = (\mathcal{V}^{IS}, \mathcal{E}^{IS})$. Each vertex in $\mathcal{G}^{IS}$ represents the prediction vector $\mathbf{z} \in \mathbb{R}^C$ for one instance within a batch, and is connected to all instances within the batch including itself. The attribute of edge $\mathcal{E}_{i,j}^{IS}$ connecting vertices $i$ and $j$ describes the class-wise relations between the predictions of instances $i$ and $j$. In practice, we can organise all edges into matrix $\mathcal{E}^{IS} \in \mathbb{R}^{B \times B \times C}$, in which:

$$\mathcal{E}_{i,j}^{IS} = \frac{\mathbf{z}_i - \mathbf{z}_j}{\|\mathbf{z}_i - \mathbf{z}_j\|_2} \in \mathbb{R}^C \quad \text{for} \quad i,j \in [1, B], \tag{3}$$

where we empirically find that normalisation along the secondary dimension helps regularise relations and improves performance (see Sec. 4.3). Concretely, $\mathcal{E}$ encodes inter-sample class-wise relations within a batch of $B$ training samples. Our method of encoding inter-sample affinity is different from and more effective than previous methods that leverage Gram matrices (Park et al., 2019; Passalis & Tefas, 2018; Tung & Mori, 2019; Huang et al., 2022) or third-order angular distances (Park et al., 2019).

### 3.4 CONSTRUCTING INTER-CLASS RELATIONS

Although $\mathcal{G}^{IS}$ densely encodes rich inter-sample relations within a batch, it fails to explicitly model inter-class patterns that are also beneficial structured knowledge (Huang et al., 2022). We propose to build and transfer a novel inter-class batch-wise relation graph $\mathcal{G}^{IC} = (\mathcal{V}^{IC}, \mathcal{E}^{IC})$, instead of Gram matrices in the inner product space in Huang et al. (2022). The construction of $\mathcal{G}^{IC}$ mirrors that of $\mathcal{G}^{IS}$. Vertices in $\mathcal{G}^{IC}$ are the class-wise logit vectors $\mathbf{w} \in \mathbb{R}^B$. Each edge in $\mathcal{E}^{IC} \in \mathbb{R}^{C \times C \times B}$ embeds the pairwise difference between the $i$-th and $j$-th per-class vectors.

$$\mathcal{E}_{i,j}^{IC} = \frac{\mathbf{w}_i - \mathbf{w}_j}{\|\mathbf{w}_i - \mathbf{w}_j\|_2} \in \mathbb{R}^B \quad \text{for} \quad i,j \in [1, C]. \tag{4}$$

Our inter-class relations preserve the batch-wise discrepancies by treating them as a dimension of additional knowledge (reciprocal to the case of inter-sample relations), which is unlike any other previous methods (Huang et al., 2022). We demonstrate in Tab. 16 that our formulation of inter-sample and inter-class relations by preserving the raw affinity knowledge along the secondary dimension performs significantly better than previous relation encoders $\psi(\cdot)$ via Gram matrices or third-order angular distances. Furthermore, We will show in Sec. 4.3 that knowledge transfer with our formulation works reasonably well with various distance metrics $\phi(\cdot)$.

### 3.5 CONSTRUCTING VIRTUAL RELATIONS

For each prediction $\mathbf{z}_i$ within a batch $\{\mathbf{z}_i\}_{i=1}^B$, we create a virtual view of it, denoted as "$\tilde{\mathbf{z}}_i$", by applying semantic-preserving transformations to original image $\mathbf{x}_i$. While other common image transformation operations are applicable, we choose RandAugment (Cubuk et al., 2020) that applies stochastic image transformations (see A.2 for details). With our batch of predictions augmented into $\{\mathbf{z}_i, \tilde{\mathbf{z}}_i\}_{i=1}^B$, we can construct a larger inter-sample edge matrix $\mathcal{E}^{IS} \in \mathbb{R}^{2B \times 2B \times C}$ and a larger inter-class edge matrix $\mathcal{E}^{IC} \in \mathbb{R}^{C \times C \times 2B}$. From the perspective of sample views, our new $\mathcal{E}^{IS}$ encompasses inter-class batch-wise prediction affinity between within-view instance predictions. Our new $\mathcal{E}^{IS}$ essentially encode three types of knowledge, namely relations amongst real views (denoted as "real-real"), relations amongst virtual views ("virtual-virtual"), and relations between

Table 1: Results for same- and different-architecture teacher-student pairs on CIFAR-100. †: using re-trained, stronger teachers.

| Teacher
Student | Venue | ResNet56
ResNet20 | ResNet32×4
ResNet8×4 | WRN-40-2
WRN-16-2 | WRN-40-2
WRN-40-1 | VGG13
VGG8 | ResNet32×4
ShuffleNetV2 | VGG13
MobileNetV2 | ResNet50
MobileNetV2 | WRN-40-2
ShuffleNetV1 |
|---|---|---|---|---|---|---|---|---|---|---|
| Teacher | | 72.34 | 79.42 | 75.61 | 75.61 | 74.64 | 79.42 | 74.64 | 79.34 | 75.61 |
| Student | | 69.06 | 72.50 | 73.26 | 71.98 | 70.36 | 71.82 | 64.60 | 64.60 | 70.50 |
| *Feature-based* | | | | | | | | | | |
| FitNets | ICLR'15 | 69.21 | 73.50 | 73.58 | 72.24 | 71.02 | 73.54 | 64.16 | 63.16 | 73.73 |
| AT | ICLR'17 | 70.55 | 73.44 | 74.08 | 72.77 | 71.43 | 72.73 | 59.40 | 58.58 | 73.32 |
| AB | AAAI'19 | 69.47 | 73.17 | 72.50 | 72.38 | 70.94 | 74.31 | 66.06 | 67.20 | 73.34 |
| OFD | ICCV'19 | 70.98 | 74.95 | 75.24 | 74.33 | 73.95 | 76.82 | 69.48 | 69.04 | 75.85 |
| VID | CVPR'19 | 70.38 | 73.09 | 74.11 | 73.30 | 71.23 | 73.40 | 65.56 | 67.57 | 73.61 |
| CRD | ICLR'20 | 71.16 | 75.51 | 75.48 | 74.14 | 73.94 | 75.65 | 69.63 | 69.11 | 76.05 |
| SRRL | ICLR'21 | 71.13 | 75.33 | 75.59 | 74.18 | 73.44 | - | - | - | - |
| PEFD | NeurIPS'22 | 70.07 | 76.08 | 76.02 | 74.92 | 74.35 | - | - | - | - |
| CAT-KD | CVPR'23 | 71.05 | 76.91 | 75.60 | 74.82 | 74.65 | 78.41 | 69.13 | 71.36 | 77.35 |
| TaT | CVPR'22 | 71.59 | 75.89 | 76.06 | 74.97 | 74.39 | - | - | - | - |
| ReviewKD | CVPR'21 | 71.89 | 75.63 | 76.12 | 75.09 | 74.84 | 77.78 | 70.37 | 69.89 | 77.14 |
| NORM | ICLR'23 | 71.35 | 76.49 | 75.65 | 74.82 | 73.95 | 78.32 | 69.38 | 71.17 | 77.63 |
| FCFD | ICLR'23 | 71.96 | 76.62 | 76.43 | 75.46 | 75.22 | 78.18 | 70.65 | 71.00 | 77.99 |
| *Logit-based* | | | | | | | | | | |
| KD | arXiv'15 | 70.66 | 73.33 | 74.92 | 73.54 | 72.98 | 74.45 | 67.37 | 67.35 | 74.83 |
| DML | CVPR'18 | 69.52 | 72.12 | 73.58 | 72.68 | 71.79 | 73.45 | 65.63 | 65.71 | 72.76 |
| TAKD | AAAI'20 | 70.83 | 73.81 | 75.12 | 73.78 | 73.23 | 74.82 | 67.91 | 68.02 | 75.34 |
| CTKD | AAAI'23 | 71.19 | 73.79 | 75.45 | 73.93 | 73.52 | 75.31 | 68.46 | 68.47 | 75.78 |
| NKD | ICCV'23 | 70.40 | 76.35 | 75.24 | 74.07 | 74.86 | 76.26 | 70.22 | 70.76 | 75.96 |
| DKD | CVPR'22 | 71.97 | 76.32 | 76.24 | 74.81 | 74.68 | 77.07 | 69.71 | 70.35 | 76.70 |
| LSKD | CVPR'24 | 71.43 | 76.62 | 76.11 | 74.37 | 74.36 | 75.56 | 68.61 | 69.02 | - |
| TTM | ICLR'24 | 71.83 | 76.17 | 76.23 | 74.32 | 74.33 | 76.55 | 69.16 | 69.59 | 75.42 |
| MLLD | CVPR'23 | 72.19 | 77.08 | 76.63 | 75.35 | 75.18 | 78.44 | 70.57 | 71.04 | 77.44 |
| TGeoKD | ICLR'24 | 72.98 | 77.27 | - | 75.43 | - | 76.89 | - | - | 77.05 |
| *Relation-based* | | | | | | | | | | |
| RKD | CVPR'19 | 69.61 | 71.90 | 73.35 | 72.22 | 71.48 | 73.21 | 64.52 | 64.43 | 72.21 |
| PKT | ECCV'18 | 70.34 | 73.64 | 74.54 | 73.45 | 72.88 | 74.69 | 67.13 | 66.52 | 73.89 |
| CCKD | CVPR'19 | 69.63 | 72.97 | 73.56 | 72.21 | 70.71 | 71.29 | 64.86 | 65.43 | 71.38 |
| SP | ICCV'19 | 69.67 | 72.94 | 73.83 | 72.43 | 72.68 | 74.56 | 66.30 | 68.08 | 74.52 |
| ICKD | ICCV'21 | 71.76 | 75.25 | 75.64 | 74.33 | 73.42 | - | - | - | - |
| DIST† | NeurIPS'22 | 71.75 | 76.31 | - | 74.73 | - | 77.35 | 68.50 | 68.66 | 76.40 |
| **VRM** | - | **72.09** | **78.76** | **77.47** | **76.46** | **76.19** | **79.34** | **71.66** | **72.30** | **78.62** |

pairs of real and virtual views ("real-virtual"). For instance, a real-virtual edge that connects real vertex $m$ and virtual vertex $n$ in $\mathcal{E}^{IS}$ is computed as $\mathcal{E}_{m,n}^{IS} = \frac{\mathbf{z}_m - \tilde{\mathbf{z}}_n}{\|\mathbf{z}_m - \tilde{\mathbf{z}}_n\|_2} \in \mathbb{R}^C$.

## 3.6 Pruning into Sparse Graphs

**Pruning redundant edges.** The augmented $\mathcal{E}^{IS}$ in Sec. 3.5 contains $2B \times 2B$ edges in dense connections, leading to quadrupled memory and computational overheads. For better efficiency, we propose to prune $\mathcal{G}^{IS}$ into sparse graphs. We begin by noticing that $\mathcal{E}^{IS}$ is symmetric along its diagonals, and prune its redundant half. This leads to up to $50\%$ reduction in the number of edges. We also remove intra-view edges as we empirically find that they are redundant and hurt knowledge transfer performance. We postulate that this is because virtual views are laden with most of the essential knowledge of the real views they are generated from, which makes the former also redundant. This step leads to a further $2\times$ edge reduction. For $\mathcal{G}^{IC}$, we decompose the augmented batch of predictions of size $2B$ into a real-view batch and a virtual-view batch, both of size $B$, and in lieu use the inter-sample batch-wise affinity vectors between them as its vertices. Compared to their original intra-view formulation of size $2B$ in Sec. 3.5, this new design encodes purely cross-view affinity knowledge with halved parameters. Our graphs now become sparse and the remaining edges can again be rearranged into compact matrices: $\mathcal{E}^{ISV} \in \mathbb{R}^{B \times B \times C}$ and $\mathcal{E}^{ICV} \in \mathbb{R}^{C \times C \times B}$, both encoding purely cross-view virtual relational knowledge. Pruning redundant edges reduces the peak GPU memory usage of our VRM module from 25.1MB to 8.93MB.

**Pruning unreliable edges.** To mitigate the diffusive effect of spurious predictions discovered in our pilot studies, we further propose to identify and prune unreliable edges. In previous graph learning works, the absolute certainty of two vertices are often used to determine the reliability of an edge. For instance, REM (Chen et al., 2020a) computes the reliability of an edge as the mean of the maximum predicted probabilities of two samples (*i.e.*, two vertices). However, we argue that this will cause the learning to be biased towards easy samples. Instead, we measure the discrepancy between two predictions. The larger this discrepancy, the more unreliable the relation constructed from them. Mathematically, the unreliable edge pruning criterion is given by:

$$\mathcal{E}_{i,j}^{ISV} = \emptyset \quad \text{if} \quad \mathrm{H}(\mathbf{z}_i^s, \tilde{\mathbf{z}}_j^s) > P_n \tag{5}$$

where $\mathrm{H}(\cdot)$ computes the joint entropy (JE) between two predictions and $P_n$ is the $n$-th percentile within the batch. While other measures of edge uncertainty may be used, JE suits our purpose with several appealing properties: 1) higher discrepancy between two vertices leads to higher JE, which is a relative measure of uncertainty; 2) as two predictions get aligned, JE approaches their individual uncertainty, which is an absolute measure of uncertainty. As such, our criterion takes account of both

relative and absolute edge uncertainties throughout the learning process. Also, note that the criterion is enforced on student predictions, whcih results in adaptive and dynamic pruning as different edges get pruned in each iteration, which improves learning.

### 3.7 FURTHER DESIGNS FOR RELAXED MATCHING

We implement further designs to mitigate the issues pinpointed in Sec. 3.2.

**Relaxed matching with logit adaptors.** A recent work (Chen et al., 2022b) argues that MLP-based adaptors improve the generalisation of features learnt in KD. Yet, their effect on relation-based KD remains unexplored. In this work, we propose to use MLP-processed student logits to construct our affinity graphs, and empirically unveil that adaptors benefit relation-based KD. In practice, we use a 1-layer MLP as relation adaptors for each student logit vector and each graph.

**Relaxed matching with logit normalisation.** Moreover, inspired by Sun et al. (2024), we apply Z-score normalisation (ZSNorm) to all logit vectors before using them to construct VRM graphs. We find ZSNorm helps relax the restrictions on the logits which in turn construct and match better relations. This is the first time ZSNorm is found useful for RM, which complements the findings by Sun et al. (2024) on IM.

These designs share similar spirits in relaxing the matching objective: We do not require the raw logits to directly produce the desired relations. Instead, we only expect a processed (*i.e.*, adapted or normalised) version of them to achieve so. Such relaxed matching alleviates overfitting, while the adaptation and normalisation operations further mitigate the effect of outliers or spurious signals.

### 3.8 FULL OBJECTIVE

With $\mathcal{G}^{IS}$ and $\mathcal{G}^{IC}$ constructed for both teacher and student predictions, our VRM objective matches $\mathcal{E}^{ISV}$ and $\mathcal{E}^{ICV}$ between teacher and student via distance metric $\phi(\cdot)$. Formally, $L_{vrm}^{ISV} = \phi(\mathcal{E}_S^{ISV}, \mathcal{E}_T^{ISV})$ and $L_{vrm}^{ICV} = \phi(\mathcal{E}_S^{ICV}, \mathcal{E}_T^{ICV})$. The full optimisation objective of VRM is a weighted combination of the CE loss and the proposed VRM losses:

$$L_{total} = L_{ce} + \alpha L_{vrm}^{ISV} + \beta L_{vrm}^{ICV} \qquad (6)$$

where $L_{ce}$ is the CE loss applied to student's predictions of both real and virtual views and supervised by GT labels; $\alpha$ and $\beta$ are scalars to balance different loss terms.

## 4 EXPERIMENTS

### 4.1 EXPERIMENTAL SETTINGS

Our method is evaluated on both image classification and object detection tasks. For image classification, we benchmark our method on CIFAR-100 (Krizhevsky, 2009) and ImageNet (Deng et al., 2009). For object detection, we experiment with the MS-COCO (Lin et al., 2014) dataset. Our experimental configurations strictly follow the standard practice in prior works. Descriptions of the datasets and more implementation details are provided in A.3.

### 4.2 MAIN RESULTS

**Results on CIFAR-100.** For *same-architecture KD*, (left of Tab. 1), VRM surpasses all previous relation-based methods across all teacher-student pairs experimented by large margins. For instance,

Table 2: Results on ImageNet. †: using retrained, stronger teachers.

| Teacher Student | Venue | ResNet34 ResNet18 | ResNet50 MobileNetV1 |
|---|---|---|---|
| Teacher | Venue | 73.31/91.42 | 76.16/92.86 |
| Student | | 69.75/89.07 | 68.87/88.76 |
| *Feature-based* | | | |
| AT | ICLR'17 | 70.69/90.01 | 69.56/89.33 |
| AB | AAAI'19 | - | 68.89/88.71 |
| OFD | ICCV'19 | 70.81/89.98 | 71.25/90.34 |
| CRD | ICLR'20 | 71.17/90.13 | 71.37/90.41 |
| CAT-KD | CVPR'23 | 71.26/90.45 | 72.24/91.13 |
| SimKD | CVPR'22 | 71.59/90.48 | 72.25/90.86 |
| ReviewKD | CVPR'21 | 71.61/90.51 | 72.56/91.00 |
| SRRL | ICLR'21 | 71.73/90.60 | 72.49/90.92 |
| PEFD | NeurIPS'22 | 71.94/90.68 | 73.16/91.24 |
| FCFD | ICLR'23 | 72.24/90.74 | 73.37/91.35 |
| *Logit-based* | | | |
| KD | arXiv'15 | 70.66/89.88 | 68.58/88.98 |
| DML | CVPR'18 | 70.82/90.02 | 71.35/90.31 |
| TAKD | AAAI'20 | 70.78/90.16 | 70.82/90.01 |
| CTKD | AAAI'23 | 71.51/- | 90.47/- |
| DKD | CVPR'22 | 71.70/90.41 | 72.05/91.05 |
| NKD | ICCV'23 | 71.96/- | 72.58/- |
| SDD | CVPR'24 | 71.14/90.05 | 72.24/90.71 |
| MLLD | CVPR'23 | 71.90/90.55 | 73.01/91.42 |
| TTM | ICLR'24 | 72.19/- | 73.09/- |
| LSKD | CVPR'24 | 72.08/90.74 | 73.22/91.59 |
| TGeoKD | ICLR'24 | 72.89/91.80 | 72.46/90.95 |
| *Relation-based* | | | |
| FSP | CVPR'17 | 70.58/89.61 | - |
| RKD | CVPR'19 | 71.34/90.37 | 71.32/90.62 |
| CCKD | CVPR'19 | 70.74/- | - |
| ICKD | ICCV'21 | 72.19/90.72 | - |
| DIST† | NeurIPS'22 | 72.07/90.42 | 73.24/91.12 |
| **VRM** | - | **72.98/91.86** | **74.04/91.73** |

Table 3: Results on MS-COCO.

| | ResNet101–ResNet18 | | | ResNet50–MobileNetV2 | | |
|---|---|---|---|---|---|---|
| | AP | AP$_{50}$ | AP$_{75}$ | AP | AP$_{50}$ | AP$_{75}$ |
| Teacher | 42.04 | 62.48 | 45.88 | 40.22 | 61.02 | 43.81 |
| Student | 33.26 | 53.61 | 35.26 | 29.47 | 48.87 | 30.90 |
| *Feature-based* | | | | | | |
| FitNets | 34.43 | 54.16 | 36.71 | 30.20 | 49.80 | 31.69 |
| FGFI | 35.44 | 55.51 | 38.17 | 31.16 | 50.68 | 32.92 |
| TAKD | 34.59 | 55.35 | 37.12 | 31.26 | 51.03 | 33.46 |
| ReviewKD | 36.75 | 56.72 | 34.00 | 33.71 | 53.15 | 36.13 |
| FCFD | 37.37 | 57.60 | 40.34 | 34.97 | 55.04 | 37.51 |
| *Logit-based* | | | | | | |
| KD | 33.97 | 54.66 | 36.62 | 30.13 | 50.28 | 31.35 |
| CTKD | 34.56 | 55.43 | 36.91 | 31.39 | 52.34 | 33.10 |
| LSKD | - | - | - | 31.74 | 52.77 | 33.40 |
| DKD | 35.05 | 56.60 | 37.54 | 32.34 | 53.77 | 34.01 |
| *Relation-based* | | | | | | |
| **VRM** | **36.67** | **57.35** | **37.71** | **32.78** | **54.19** | **34.25** |

Table 4: Results for ConvNet-to-ViT distillation on CIFAR-100. Hier.: hierarchical structure.

| Student | Hier. | Size | Baseline | KD | AT | SP | LG | AutoKD | LSKD | VRM |
|---|---|---|---|---|---|---|---|---|---|---|
| DeiT-Ti | ✗ | 5M | 65.08 | 73.25 | 73.51 | 67.36 | 78.15 | 78.58 | 78.55 | **79.19** |
| T2T-ViT-7 | ✗ | 4M | 69.37 | 74.15 | 74.01 | 72.26 | 78.35 | 78.62 | 78.43 | **78.83** |
| PiT-Ti | ✓ | 5M | 73.58 | 75.47 | 76.03 | 74.97 | 78.48 | 78.51 | 78.76 | **79.25** |
| PVT-Ti | ✓ | 13M | 69.22 | 74.66 | 77.07 | 70.48 | 77.48 | 73.60 | 78.43 | **79.42** |

Table 5: Ablation of major design choices in VRM.

| Component | ResNet32×4
ResNet8×4 | VGG13
VGG8 |
|---|---|---|
| Baseline | 72.12 | 69.67 |
| + Match $\mathcal{E}^{IS}$ | 75.64 | 74.08 |
| + Match $\mathcal{E}^{IC}$ | 75.94 | 74.47 |
| + Match $\{\mathcal{E}^{ISV}, \mathcal{E}^{ICV}\}$ | 77.13 | 75.35 |
| + ZSNorm of $\mathcal{V}$ | 78.06 | 75.57 |
| + Probs as $\mathcal{V}$ | 78.47 | 75.67 |
| + L2Norm of $\{\mathcal{E}^{ISV}, \mathcal{E}^{ICV}\}$ | 78.68 | 75.75 |
| + JE Pruning | **78.76** | **76.19** |

Table 6: Effect of different relation distance metrics.

| Distance
Metric | ResNet32×4
ResNet8×4 | VGG13
VGG8 |
|---|---|---|
| L2 | 78.54 | 75.21 |
| L1 | 77.97 | 75.62 |
| Huber | **78.76** | **76.19** |
| Cosine | 78.62 | 75.50 |
| Pearson | 78.23 | 75.73 |
| KLD | 78.39 | 75.28 |
| MMD | 77.91 | 75.67 |

Table 7: Effect of different $\alpha$, $\beta$, $\tau$, and $n$.

| $\alpha$ | 32 | 64 | 128 | 256 | 512 |
|---|---|---|---|---|---|
| Acc. (%) | 78.03 | 78.13 | **78.76** | 78.32 | 77.84 |

| $\beta$ | 8 | 16 | 32 | 64 | 128 |
|---|---|---|---|---|---|
| Acc. (%) | 78.40 | 78.51 | **78.76** | 78.86 | 78.60 |

| $\tau$ | 1 | 2 | 3 | 4 | 5 | 6 |
|---|---|---|---|---|---|---|
| Acc. (%) | 78.16 | 78.69 | 78.57 | **78.76** | 78.41 | 78.63 |

| $n$ | 0 | 1 | 2 | 3 | 4 | 5 |
|---|---|---|---|---|---|---|
| Acc. (%) | 77.92 | 78.38 | **78.76** | 78.55 | 78.43 | 78.50 |

it outscores current best-performer DIST by 2.45% and 1.73% for ResNet32×4 →ResNet8×4 and WRN-40-2→WRN-40-1, respectively. Noticeably, VRM is able to significantly outperform even the strongest feature-based method, FCFD. On average, it also performs much better than top logit-based methods such as TGeoKD and MLLD. These results are significant, given that IM methods are known to be naturally more adapt at homogeneous pairs. For *different-architecture KD*, which RM methods are supposed to be more competent with, VRM readily surpasses *all* existing RM and IM methods with notable margins. These results reveal the marked versatility of VRM on both homogeneous and heterogeneous teacher-student pairs. More results are found in A.4.

**Results on ImageNet.** VRM surpasses all feature-based and relation-based methods on the large-scale ImageNet dataset, shown in Tab. 2. Akin to findings on CIFAR-100, the advantage of VRM is more apparent over the heterogeneous pair of ResNet50→MobileNetV2, whereby it produces the strongest performance and for the first time hits 74.0% Top-1 accuracy. Notably, VRM outperforms strong competitors such as FCFD, PEFD, and NORM with comparable or even less computational overheads as compared in Tab. 9.

**Results on MS-COCO.** We demonstrate that VRM generalises to more challenging tasks by adapting it to object detection. As shown in Tab. 3, VRM delivers better performance than existing logit-based methods and competitive results to feature-based methods. VRM is slightly behind top-performing feature-based methods such as FCFD and ReviewKD. This is a consequence of the very nature of the object detection task, where fine-grained contextual features play a vital role, making feature-based methods inherently better-off (Wang et al., 2019). Nonetheless, we experimentally demonstrate that VRM is effective on object detection; VRM improves the vanilla KD baseline by 2.70% AP, surpasses all logit-based methods, and is on par with strong feature-based methods.

**Results for ConvNet-to-ViT Distillation** We further validate VRM on the task of ConvNet-to-ViT distillation. To do so, we train a ResNet56 teacher on CIFAR-100 and distill its knowledge into different ViT-based students, namely DeiT (Touvron et al., 2021), T2T-ViT (Yuan et al., 2021), PiT (Heo et al., 2021), and PVT (Wang et al., 2021). Tab. 4 suggests that VRM is highly effective under this setting, producing the highest results for different ViT architectures. For example, by simply replacing the vanilla KD objective with our VRM and making no other modifications, VRM improves DeiT-Ti by 5.94%. VRM leads RM-based SP by 11.83% for DeiT-Ti and even surpasses AutoKD (Li et al., 2023a) with AutoML-based search. Detailed configurations are found in A.3.

### 4.3 Ablation Studies

**Ablations of main design choices.** We first perform ablations on our main design choices. We start from the baseline where only the CE loss is applied, and gradually incorporate our designs described in previous sections. As shown in Tab. 5, each extra design consistently brings performance gains, which corroborate the validity of our individual design choices.

**Choice of distance metrics for relation matching.** We investigate alternative choices of distance metric in matching the proposed affinity graphs $\mathcal{E}^{ISV}$ and $\mathcal{E}^{ICV}$. From Tab. 6, we observe that our VRM objective works reasonably well with most distance metrics studied.

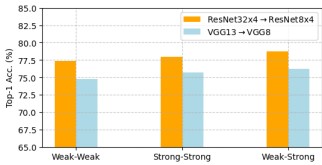 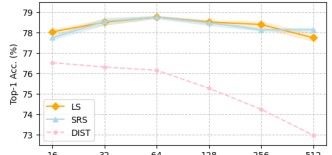 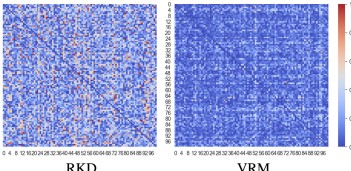

Figure 3: Effect of different difficulty pairs.

Figure 4: Robustness to varying batch sizes.

Figure 5: Teacher-student prediction discrepancy maps.

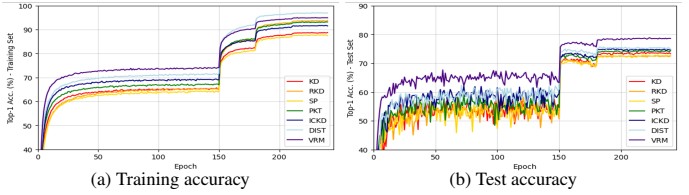

(a) Training accuracy    (b) Test accuracy

Figure 6: Visualisation of training dynamics.

Table 8: Effect of matching graph vertices.

| Teacher
Student | ResNet32×4
ResNet8×4 | VGG13
VGG8 | ResNet50
MobileNetV2 |
|---|---|---|---|
| Matching $\{\mathcal{E}\}$ | **78.76** | **76.19** | **72.30** |
| Matching $\{\mathcal{E}, \mathcal{V}\}$ | 78.63 | 75.60 | 70.97 |
| Matching $\{\mathcal{V}\}$ | 78.46 | 75.77 | 68.33 |

**Effect of varying hyperparameters.** VRM involves three major hyperparameters in its objective formulation: $\alpha$, $\beta$, and $\tau$. According to Tab. 7, VRM works generally well with $\alpha$ and $\beta$ in a reasonable range. Larger $\alpha$ and $\beta$ may produce better results for certain KD pairs but worse results for others. Hence, we default $\alpha$ to 128 and $\beta$ to 32 in favour of generalisation. For $\tau$, we simply opt for the common choice of $\tau = 4$ (Hinton et al., 2015) and find that it produces the best results.

**Effect of different real-virtual difficulty gaps.** We probe into the effect of different difficulty gaps in our real-virtual relation matching formulation. We first tune RandAugment's parameter $n$, which controls the number of transformations randomly applied to create the virtual view. The larger the $n$, the more difficult the image becomes and the larger discrepancy between the logits of both views. From Tab. 7, $n = 2$ gives the best results, whereas other values also report competitive results compared to state-of-the-arts. We also experiment with different strength pairs in Fig. 3 and find that cross-strength matching performs best. Overall, we conclude that 1) a moderate difficulty gap leads to optimal performance, and 2) difficulty gaps are significant to the success of VRM.

**Robustness to varying batch sizes.** Inter-sample relational methods are known to be sensitive to $B$, the number of samples within a training batch. We examine how robust VRM is against varying $B$. To adjust the learning rate accordingly, we consider two LR scaling rules: linear LR scaling (Goyal, 2017) and square root LR scaling (Chen et al., 2020b). From Fig. 4, VRM yields competitive results across a wide range of $B$ values and is therefore adequately robust to varying batch sizes. In contrast, the performance of DIST (Huang et al., 2022) deteriorates significantly as $B$ varies.

**More ablation studies.** More ablations on the effects of redundant edge pruning, different relation encoding functions, and different GT supervision policies are provided in A.5.

## 4.4 FURTHER ANALYSIS

**Visualisation of teacher & student prediction discrepancy.** We compute the mean discrepancy (Manhattan distance used) between teacher's averaged prediction distribution for each class and that of student on the CIFAR-100 validation set. From Fig. 5, VRM pulls student's predictions significantly closer to teacher's compared to RKD, while both methods do not involve direct IM matching objectives. This can be attributed to our designs such as cross-view regularisation and graph pruning that substantially improve generalisation.

**Visualisation of embedding space.** We conduct t-SNE analysis on student penultimate layer embeddings learnt via different methods. As presented in Fig. 7, VRM leads to more compact per-class clusters with clearer inter-class separation and less stray points. These imply our method induces better features in student for the downstream task. More visualisations are presented in Fig. 9.

**Analysis of training dynamics.** Figs. 6a and 6b plot the per-epoch training and validation set accuracies throughout training. VRM maintains a all-time lead in both training and validation performance, with faster convergence. Besides, while VRM's lead in training performance tapers off towards the end of training, it remains more substantial, if not further enlarging, in validation per-

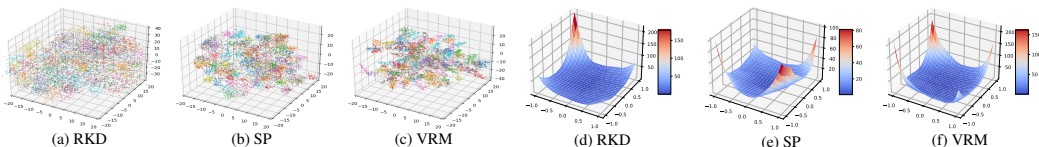

Figure 7: t-SNE (a-c) and loss landscape (d-f) visualisations for ResNet8×4 students distilled with a ResNet32×4 teacher on CIFAR-100 via different methods.

Table 9: Training efficiency of different distillation methods.

| Method | KD | RKD | ICKD | DIST | CRD | ReviewKD | SDD | LSKD | MLLD | NORM | PEFD | FCFD | VRM |
|---|---|---|---|---|---|---|---|---|---|---|---|---|---|
| Train. Time (ms) | 24.9 | 31.0 | 28.0 | 27.2 | 41.2 | 39.9 | 34.2 | 27.1 | 57.2 | 35.1 | 36.2 | 56.4 | **47.2** |
| Peak GPU Mem. Usage (MB) | 323 | 330 | 381 | 330 | 1418 | 1042 | 690 | 330 | 576 | 1806 | 701 | 953 | **579** |

formance, which highlights its superior generalisation properties. This analysis also shows that our designs are effective in mitigating the issues with existing RM methods identified in the pilot studies.

**Analysis of loss landscape.** We further analyse the generalisation and convergence properties of our method through the lens of visualised loss landscape (Li et al., 2018). From Fig. 7, VRM has the widest and flattest region of minima which is a typical hint of better model generalisation and robustness; this wide convexity basin is surrounded by salient pikes in different directions, indicative of excellent convergence properties. Loss landscapes for more methods are presented in Fig. 10.

**Training efficiency.** Tab. 9 benchmarks the training time per batch and the peak GPU memory usage of various methods on a workstation equipped with 20 Intel Core i9-10850K CPUs (10 cores) and an NVIDIA RTX 3090 GPU. All measurements are taken on CIFAR-100 with a batch size of 64. As seen, VRM's training speed and GPU memory usage are both within a reasonable range compared with existing algorithms. Notably, VRM is more efficient that top-performing feature-based (Liu et al., 2023b; Chen et al., 2022b; Liu et al., 2023a) and logit-based (Wei et al., 2024; Jin et al., 2023) methods. Our method does not introduce any additional overheads at inference.

**Vertex matching.** VRM matches edges $\mathcal{E}^{ISV}$ and $\mathcal{E}^{ICV}$ which carry relational knowledge. By extension, we can naturally expect vertices $\mathcal{V}$ to be also transferred. In Tab. 8, it is observed that matching vertices is not as effective. The advantage of matching relations (*i.e.*, edges) is more pronounced for heterogeneous KD pairs such as ResNet50→MobileNetV2. Adding vertex matching to the proposed edge matching does not improve but instead degrade performance. We argue that this is because introducing vertex matching objectives make the matching criterion more stringent which is against our motivation of using more slackened matching.

**Role of transformation operations.** We emphasise the *strong performance of VRM is not a simple outcome of the transformations* involved. This can be verified by the results in Fig. 3 where using only the default weak transformations in existing methods (*i.e.*, random crop and horizontal flip) for both views achieves 77.38% accuracy (denoted as "Weak-Weak"), compared to the baseline of 75.64% (Tab. 5) and 71.90% of RKD with exactly the same transformations. In fact, in the case of "Weak-Weak", both views still differ because of the stochastic operations used. The *success of VRM lies exactly in this discrepancy*, where cross-view regularisation comes into crucial play.

**More analyses.** More discussions, including the effect of longer training, applying VRM to features, and comparisons with existing works, are provided in A.6.

## 5 CONCLUSION

In this paper, we have presented VRM, a novel knowledge distillation framework that constructs and transfers virtual relations. Our designs are motivated by a set of pilot experiments, from which we identified two main cruxes with existing relation-based KD methods: their tendency to overfit and susceptibility to adverse gradient propagation. A series of tailored designs are developed and are shown to successfully mitigate these issues. We have conducted extensive experiments on different tasks and multiple datasets and verified VRM's validity and superiority in diverse settings, whereby VRM consistently demonstrates state-of-the-art performance. We hope that this work could renew the community's interest in relation-based knowledge distillation, and encourage more systematic reassessment of the design principles of such solutions.

REPRODUCIBILITY STATEMENT

All source code and models needed to reproduce the experiments in this paper will be made public with detailed instructions. The configurations of all experiments have also been thoroughly introduced in the main text and supplementary document of this paper. When developing our method, we also ensured that we maximally kept the default configurations of the codebase we use and previous methods we compare with.

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

# A APPENDIX

## A.1 LIST OF ALL COMPARED METHODS

A list of all methods we have compared with in this paper is as follows:

**Feature-based** methods include FitNets (Romero et al., 2015), AT (Zagoruyko & Komodakis, 2017), AB (Heo et al., 2019b), OFD (Heo et al., 2019a), VID (Ahn et al., 2019), CRD (Tian et al., 2020), SRRL (Yang et al., 2021b), SemCKD (Chen et al., 2021a), PEFD (Chen et al., 2022b), MGD (Yang et al., 2022b), CAT-KD (Guo et al., 2023), TaT (Lin et al., 2022), ReviewKD (Chen et al., 2021b), NORM (Liu et al., 2023b), and FCFD (Liu et al., 2023a).

**Logit-based** methods include KD (Hinton et al., 2015), DML (Zhang et al., 2018b), TAKD (Mirzadeh et al., 2020), CTKD (Li et al., 2023b), NKD (Yang et al., 2023b), DKD (Zhao et al., 2022), LSKD (Sun et al., 2024), TTM (Zheng & Yang, 2024), MLLD (Jin et al., 2023), SDD Wei et al. (2024), and TGeoKD (Hu et al., 2024).

**Relation-based** methods include FSP (Yim et al., 2017), RKD (Park et al., 2019), PKT (Passalis & Tefas, 2018), CCKD (Peng et al., 2019), SP (Tung & Mori, 2019), ICKD (Liu et al., 2021a), and DIST (Huang et al., 2022).

For MS-COCO object detection, we also compare VRM with FGFI (Wang et al., 2019). For ConvNet-to-ViT experiments, we also present the results for LG Li et al. (2022) and AutoKD Li et al. (2023a).

## A.2 LIST OF ALL TRANSFORMATION OPERATIONS

For our main experiments, we borrow the RandAugment implementation from the `TorchSSL` codebase [1]. It comprises a total of 14 image transformation operations, namely:

1. `Autocontrast`: automatically adjust image contrast
2. `Brightness`: adjust image brightness
3. `Color`: adjust image colour balance
4. `Contrast`: adjust image contrast
5. `Equalize`: equalise image histogram
6. `Identity`: leave image unaltered
7. `Posterize`: reduce number of bits for each channel
8. `Rotate`: rotate image
9. `Sharpness`: adjust image sharpness
10. `Shear_x`: shear image horizontally
11. `Shear_y`: shear image vertically
12. `Solarize`: invert all pixels above a threshold
13. `Translate_x`: translate image horizontally
14. `Translate_y`: translate image vertically

Besides, we also apply `Cutout` with a probability of 1.0, which sets a square patch of random size within the image to gray. The above operations are preceded by `RandomCrop` and `RandomHorizontalFlip` in our strong view image generatino pipeline.

For ConvNet-to-ViT experiments, we follow Li et al. (2022) and use the RandAugment function provided by the `timm` library [2]. This function contains 15 image transformation operations:

1. `AutoContrast`: automatically adjust image contrast
2. `Brightness`: adjust image brightness
3. `Color`: adjust image colour balance

---

[1] https://github.com/TorchSSL
[2] https://github.com/huggingface/pytorch-image-models

4. `Contrast`: adjust image contrast

5. `Equalize`: equalise image histogram

6. `Invert`: invert image

7. `Posterize`: reduce number of bits for each channel

8. `Rotate`: rotate image

9. `Sharpness`: adjust image sharpness

10. `ShearX`: shear image horizontally

11. `ShearY`: shear image vertically

12. `Solarize`: invert all pixels above a threshold

13. `SolarizeAdd`: add a certain value to all pixels below a threshold

14. `TranslateXRel`: translate image horizontally by a fraction of its width

15. `TranslateYRel`: translate image vertically by a fraction of its height

Similar to the role of `Cutout`, the `timm` library additionally implements a `RandomErasing` operation, which sets a rectangular patch of random size and shape within the image to random pixels. The above operations are preceded by `RandomResizedCropAndInterpolation` and `RandomHorizontalFlip` in our strong view image generatino pipeline, which is the default configuration in `timm`.

### A.3 DETAILS ON EXPERIMENTAL CONFIGURATIONS

**Datasets.** We conduct experiments on CIFAR-100 and ImageNet for image classification, and MS-COCO for object detection. CIFAR-100 (Krizhevsky, 2009) contains 60k $32 \times 32$ RGB images annotated in 100 classes. It is split into 50,000 training and 10,000 validation images. ImageNet (Deng et al., 2009) is a 1,000-category large-scale image recognition dataset. It provides 1.28 million RGB images for training and 5k for validation. MS-COCO (Lin et al., 2014) is an object detection dataset with images of common objects in 80 categories. We experiment with its `train2017` and `val2017` that include 118k training and 5k validation images, respectively.

**Configurations.** For CIFAR-100 main experiments, we strictly follow the standard training configurations in previous works (Liu et al., 2023a; Zhao et al., 2022; Sun et al., 2024). Specifically, we train our framework for 240 epochs with the SGD optimiser and a batch size of 64. The initial LR is 0.01 for MobileNets (Howard et al., 2017) and ShuffleNets (Zhang et al., 2018a) and 0.05 for other architectures, which decay by a factor of 10 at [150th, 180th, 210th] epochs. The momentum is set to 0.9 and weight decay to 5e-4. Softmax temperature is set to 4. For ConvNet-to-ViT experiments on CIFAR-100, our settings follow Li et al. (2023a); Sun et al. (2024).

For ImageNet experiments, same as standard practice, we train our framework for 100 epochs with a batch size of 256 on two GPUs, with an initial LR of 0.1 that decays by a factor of 10 at [30th, 60th, 90th] epochs. Moment and weight decay are set to 0.9 and 1e-4, respectively. Softmax temperature is set to 2.

For MS-COCO object detection, we adopt the configurations of Wang et al. (2019); Chen et al. (2021b); Zhao et al. (2022); Sun et al. (2024); Liu et al. (2023b) whereby we experiment with Faster-RCNN-FPN (Lin et al., 2017) with different backbone models. All models are trained for 180,000 iterations on 2 GPUs with a batch size of 8. The LR is initially set as 0.01 and decays at the 120,000th and 160,000th iterations.

**Implementations.** Our method is implemented in the `mdistiller` [3] codebase in PyTorch for image classification experiments. For object detection, it also partially builds upon the `detectron2` [4] library. For ConvNet-to-ViT experiments, we utilise the `pycls` [5] and the `tiny-transformer` [6] codebases. All reported results are average over 3 trials.

**ConvNet-to-ViT Experiments.** As few studies have considered this setting, we developed our experiments following (Sun et al., 2024; Li et al., 2023a) on the codebase provided by Li et al.

---

[3] https://github.com/megvii-research/mdistiller

[4] https://github.com/facebookresearch/detectron2

[5] https://github.com/facebookresearch/pycls

[6] https://github.com/lkhl/tiny-transformers

Table 10: Top-1 accuracy (%) on CIFAR-100 for same-architecture teacher-student pairs.

| Teacher
Student | Venue | ResNet56
ResNet20 | ResNet110
ResNet32 | ResNet32×4
ResNet8×4 | WRN-40-2
WRN-16-2 | WRN-40-2
WRN-40-1 | VGG13
VGG8 |
|---|---|---|---|---|---|---|---|
| Teacher | | 72.34 | 74.31 | 79.42 | 75.61 | 75.61 | 74.64 |
| Student | | 69.06 | 71.14 | 72.50 | 73.26 | 71.98 | 70.36 |
| *Feature-based* | | | | | | | |
| FitNets | ICLR'15 | 69.21 | 71.06 | 73.50 | 73.58 | 72.24 | 71.02 |
| AT | ICLR'17 | 70.55 | 72.31 | 73.44 | 74.08 | 72.77 | 71.43 |
| AB | AAAI'19 | 69.47 | 70.98 | 73.17 | 72.50 | 72.38 | 70.94 |
| OFD | ICCV'19 | 70.98 | 73.23 | 74.95 | 75.24 | 74.33 | 73.95 |
| VID | CVPR'19 | 70.38 | 72.61 | 73.09 | 74.11 | 73.30 | 71.23 |
| CRD | ICLR'20 | 71.16 | 73.48 | 75.51 | 75.48 | 74.14 | 73.94 |
| SRRL | ICLR'21 | 71.13 | 73.48 | 75.33 | 75.59 | 74.18 | 73.44 |
| PEFD | NeurIPS'22 | 70.07 | 73.26 | 76.08 | 76.02 | 74.92 | 74.35 |
| CAT-KD | CVPR'23 | 71.05 | 73.62 | 76.91 | 75.60 | 74.82 | 74.65 |
| TaT | CVPR'22 | 71.59 | 74.05 | 75.89 | 76.06 | 74.97 | 74.39 |
| ReviewKD | CVPR'21 | 71.89 | 73.89 | 75.63 | 76.12 | 75.09 | 74.84 |
| NORM | ICLR'23 | 71.35 | 73.67 | 76.49 | 75.65 | 74.82 | 73.95 |
| FCFD | ICLR'23 | 71.96 | - | 76.62 | 76.43 | 75.46 | 75.22 |
| *Logit-based* | | | | | | | |
| KD | arXiv'15 | 70.66 | 73.08 | 73.33 | 74.92 | 73.54 | 72.98 |
| DML | CVPR'18 | 69.52 | 72.03 | 72.12 | 73.58 | 72.68 | 71.79 |
| TAKD | AAAI'20 | 70.83 | 73.37 | 73.81 | 75.12 | 73.78 | 73.23 |
| CTKD | AAAI'23 | 71.19 | 73.52 | 73.79 | 75.45 | 73.93 | 73.52 |
| NKD | ICCV'23 | 70.40 | 72.77 | 76.35 | 75.24 | 74.07 | 74.86 |
| DKD | CVPR'22 | 71.97 | 74.11 | 76.32 | 76.24 | 74.81 | 74.68 |
| LSKD | CVPR'24 | 71.43 | 74.17 | 76.62 | 76.11 | 74.37 | 74.36 |
| TTM | ICLR'24 | 71.83 | 73.97 | 76.17 | 76.23 | 74.32 | 74.33 |
| MLLD | CVPR'23 | 72.19 | 74.11 | 77.08 | 76.63 | 75.35 | 75.18 |
| TGeoKD | ICLR'24 | 72.98 | 75.09 | 77.27 | - | 75.43 | - |
| *Relation-based* | | | | | | | |
| FSP | CVPR'17 | 69.95 | 71.89 | 72.62 | 72.91 | - | 70.20 |
| RKD | CVPR'19 | 69.61 | 71.82 | 71.90 | 73.35 | 72.22 | 71.48 |
| PKT | ECCV'18 | 70.34 | 72.61 | 73.64 | 74.54 | 73.45 | 72.88 |
| CCKD | CVPR'19 | 69.63 | 71.48 | 72.97 | 73.56 | 72.21 | 70.71 |
| SP | ICCV'19 | 69.67 | 72.69 | 72.94 | 73.83 | 72.43 | 72.68 |
| ICKD | ICCV'21 | 71.76 | 73.89 | 75.25 | 75.64 | 74.33 | 73.42 |
| DIST | NeurIPS'22 | 71.75 | - | 76.31 | - | 74.73 | - |
| **VRM** | - | **72.09** | **75.03** | **78.76** | **77.47** | **76.46** | **76.19** |

(2022). Our experimental configurations follow Li et al. (2022) and Liu et al. (2021b). Specifically, the ResNet56 teacher is trained for 300 epochs with an initial LR of 0.1 and a cosine LR schedule. The resulted pretrained teacher has a top-1 accuracy of 71.61%. All ViTs are trained for 300 epochs (including 20-epoch linear warm-up) using the AdamW optimiser. The initial LR is 5e-4 with a weight decay of 0.05, which eventually decays to 5e-6 via a cosine LR policy. The ResNet56 teacher is trained on $32 \times 32$ resolution images, while ViT students are fed with $224 \times 224$ images. The default RandAugment is applied for data augmentation, with number of randomly sampled operations $n$ set to 2, transform magnitude $m$ to 9, and probability of applying random erasing $p$ to 0.25. All models are trained on a single NVIDIA RTX 3090 GPU with a batch size of 128.

## A.4    MORE EXPERIMENTAL RESULTS

We present the full results on CIFAR-100 in Tabs. 10 and  11 for more same-architecture and different-architecture distillation pairs and for additional comparing methods.

## A.5    MORE ABLATION STUDIES

**Effect of pruning redundant edges.** We conduct ablation experiments to see the effect of pruning redundant edges, described in Sec. 3.6. As presented in Tab. 15, matching the raw and bulky inter-sample affinity graph with redundancy and duplication is not only less efficient but also inferior in terms of performance. We postulate that this is partially ascribed the fact that each vertex in the raw graph is connected to a larger and more complex set of other vertices that involve both real and virtual vertices. This complicates the learning while making each vertex more vulnerable to an increased likelihood of adverse gradient propagation. Another possible reason is that matching real-virtual cross-view relations are more regularised, as opposed to matching real-real or virtual-virtual intra-view relations that are easier and more readily overfitted. Note that we also conjectured that the degraded performance of matching the raw graph may be ascribed to different distribution patterns of the prediction vectors at the vertices since they now have a dimension of $2 \times B$ compared to $B$.

Table 11: Top-1 accuracy (%) on CIFAR-100 for different-architecture teacher-student pairs.

| Teacher
Student | Venue | ResNet32×4
ShuffleNetV2 | VGG13
MobileNetV2 | ResNet50
MobileNetV2 | ResNet50
VGG8 | ResNet32×4
ShuffleNetV1 | WRN-40-2
ShuffleNetV1 |
|---|---|---|---|---|---|---|---|
| Teacher | | 79.42 | 74.64 | 79.34 | 79.34 | 79.42 | 75.61 |
| Student | | 71.82 | 64.60 | 64.60 | 70.36 | 70.50 | 70.50 |
| *Feature-based* | | | | | | | |
| FitNets | ICLR'15 | 73.54 | 64.16 | 63.16 | 70.69 | 73.59 | 73.73 |
| AB | AAAI'19 | 74.31 | 66.06 | 67.20 | 70.65 | 73.55 | 73.34 |
| AT | ICLR'17 | 72.73 | 59.40 | 58.58 | 71.84 | 71.73 | 73.32 |
| VID | CVPR'19 | 73.40 | 65.56 | 67.57 | 70.30 | 73.38 | 73.61 |
| OFD | ICCV'19 | 76.82 | 69.48 | 69.04 | - | 75.98 | 75.85 |
| CRD | ICLR'20 | 75.65 | 69.63 | 69.11 | 74.30 | 75.11 | 76.05 |
| MGD | ECCV'22 | 76.65 | 69.44 | 68.54 | 73.89 | 76.22 | 75.89 |
| SemCKD | AAAI'21 | 77.02 | 69.98 | 68.69 | 74.18 | 76.31 | 76.06 |
| ReviewKD | CVPR'21 | 77.78 | 70.37 | 69.89 | 75.34 | 77.45 | 77.14 |
| NORM | ICLR'23 | 78.32 | 69.38 | 71.17 | 75.67 | 77.79 | 77.63 |
| FCFD | ICLR'23 | 78.18 | 70.65 | 71.00 | - | 78.12 | 77.99 |
| CAT-KD | CVPR'23 | 78.41 | 69.13 | 71.36 | - | 78.26 | 77.35 |
| *Logit-based* | | | | | | | |
| KD | arXiv'15 | 74.45 | 67.37 | 67.35 | 73.81 | 74.07 | 74.83 |
| DML | CVPR'18 | 73.45 | 65.63 | 65.71 | - | 72.89 | 72.76 |
| TAKD | AAAI'20 | 74.82 | 67.91 | 68.02 | - | 74.53 | 75.34 |
| CTKD | AAAI'23 | 75.31 | 68.46 | 68.47 | - | 74.48 | 75.78 |
| NKD | ICCV'23 | 76.26 | 70.22 | 70.76 | 74.01 | 75.31 | 75.96 |
| DKD | CVPR'22 | 77.07 | 69.71 | 70.35 | - | 76.45 | 76.70 |
| LSKD | CVPR'24 | 75.56 | 68.61 | 69.02 | - | - | - |
| TTM | ICLR'24 | 76.55 | 69.16 | 69.59 | 74.82 | 74.37 | 75.42 |
| TGeoKD | ICLR'24 | 76.89 | - | - | - | 76.83 | 77.05 |
| SDD | CVPR'24 | 76.67 | 68.79 | 69.55 | 74.89 | 76.30 | 76.54 |
| MLLD | CVPR'23 | 78.44 | 70.57 | 71.04 | - | 77.18 | 77.44 |
| CRLD | MM'24 | 78.27 | 70.37 | 70.39 | 71.36 | - | - |
| CRLD | MM'24 | 78.27 | 70.37 | 70.39 | 71.36 | - | - |
| *Relation-based* | | | | | | | |
| RKD | CVPR'19 | 73.21 | 64.52 | 64.43 | 71.50 | 72.28 | 72.21 |
| PKT | ECCV'18 | 74.69 | 67.13 | 66.52 | 73.01 | 74.10 | 73.89 |
| CCKD | CVPR'19 | 71.29 | 64.86 | 65.43 | 70.25 | 71.14 | 71.38 |
| SP | ICCV'19 | 74.56 | 66.30 | 68.08 | 73.34 | 73.48 | 74.52 |
| DIST | NeurIPS'22 | 77.35 | 68.50 | 68.66 | 74.11 | 76.34 | 76.40 |
| **VRM** | - | **79.34** | **71.66** | **72.30** | **76.96** | **78.28** | **78.62** |

Table 12: Effect of $L_{ce}^V$ supervision.

| | ResNet32×4
ResNet8×4 | VGG13
VGG8 |
|---|---|---|
| Baseline | **78.76** | **76.19** |
| w/o $L_{ce}^S$ | 78.47 | 75.66 |

Table 13: Effect of longer training.

| | KD | RKD | DIST | VRM |
|---|---|---|---|---|
| Baseline | 73.83 | 72.63 | 76.16 | 78.76 |
| LT | 73.82 | 72.49 | 75.90 | **78.97** |

Table 14: Applying VRM to network features.

| Method | Location | ResNet32×4
ResNet8×4 | VGG13
VGG8 |
|---|---|---|---|
| RKD | | 72.63 | 70.87 |
| PKT | | 74.41 | 72.78 |
| CRD | pooled_feats | 75.51 | 73.94 |
| ReviewKD | | 75.63 | 74.84 |
| VRM | | **76.39** | **74.92** |
| VRM | logits | **78.76** | **76.19** |

We experimented with different temperature $\tau$ in an attempt to re-adjust the distributions to be more relation-matching-friendly, but the results remain inferior.

**Effect of different relation encoding functions.** VRM introduces a novel relation encoding function that models the inter-correlations in the primary dimension while preserving raw knowledge along the secondary dimension. This translates into modelling inter-sample relations while preserving pairwise class-wise distance as auxiliary knowledge for our $\mathcal{E}^{ISV}$, and modelling inter-class relations while preserving pairwise instance-wise distance as auxiliary knowledge for $\mathcal{E}^{ICV}$. As a result, our relation matrices have shape $[B, B, C]$ for $\mathcal{E}^{ISV}$ and $[C, C, B]$ for $\mathcal{E}^{ICV}$, which are different from those used by previous relation-based methods, as listed in Tab. 16. To demonstrate the superiority of our formulation, we substitute the proposed relation encoding functions with existing Gram matrices (Peng et al., 2019; Tung & Mori, 2019; Passalis & Tefas, 2018; Huang et al., 2022) or angle-wise relations (Park et al., 2019) and find that ours yield significantly better performance, as shown in Tab. 16. These ablation results validate the superiority of VRM's formulation of relations.

**Effect of GT supervision policies.** Tab. 12 shows the effect of removing the GT supervision on the student model's predictions of the virtual-view image. It demonstrates that supervising student predictions of the virtual view is important for ensuring the quality of the virtual view predictions. The quality of vertices have a direct impact on the quality of the edges (*i.e.*, relations) constructed

Table 15: Effect of pruning redundant edges.

| | ResNet32×4 ResNet8×4 | VGG13 VGG8 |
|---|---|---|
| Redundant $\mathcal{E}$ ($\tau = 1$) | 77.39 | 75.26 |
| Redundant $\mathcal{E}$ ($\tau = 2$) | 77.80 | 74.94 |
| Redundant $\mathcal{E}$ ($\tau = 4$) | 77.19 | 75.30 |
| Pruned $\mathcal{E}$ ($\tau = 4$) | **78.76** | **76.19** |

Table 16: Effect of different relation encoding function $\psi(\cdot)$.

| $\psi(\cdot)$ | Shape of Rel. Matrix | Method | ResNet32×4 ResNet8×4 | VGG13 VGG8 | ResNet50 MobileNetV2 |
|---|---|---|---|---|---|
| Gram matrices | $[B, B]$ & $[C, C]$ | CCKD, SP, PKT, DIST | 77.31 | 75.12 | 70.28 |
| Angle-wise relations | $[B, B, B]$ | RKD | 77.14 | 75.09 | 69.65 |
| Ours | $[B, B, C]$ & $[C, C, B]$ | VRM | **78.76** | **76.19** | **72.30** |

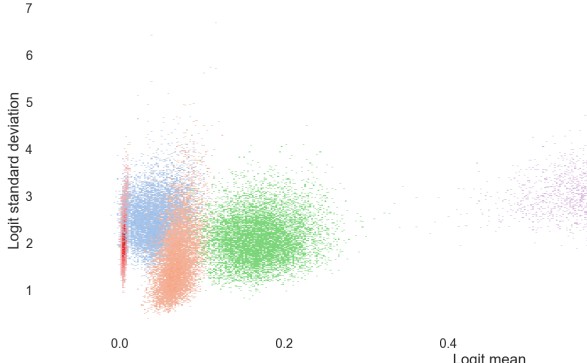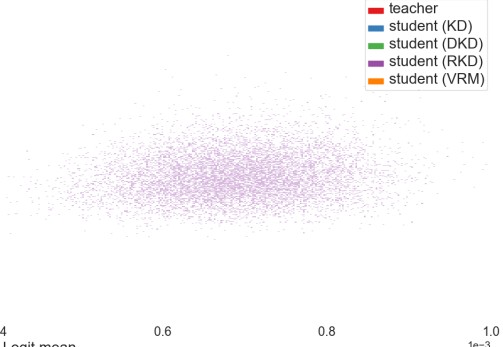

Figure 8: Bivariate histogram of the mean and standard deviation of logits predicted by different models on CIFAR-100.

within the affinity graphs. As such, we choose to also supervise the virtual vertices of our graphs with GT labels.

### A.6 MORE ANALYSES

**Analysis of logit mean & standard deviation.** In Fig. 8, we plot the histogram of the mean and standard deviation of instance-wise logit predictions given by various method. IM methods are found to produce logits closer to the teachers' in terms of both logit mean and standard deviation distributions. Intriguingly, the proposed VRM, being a purely relation-based method that is free of any explicit instance-wise logit matching, is on par with IM methods in this regard and markedly outdoes DKD. This suggests that the proposed cross-view relational matching provides strong regularisation that better enables the student to learn the underlying logit distribution of the teacher. This is particularly evident given that RKD, a relation-based method which also has an IM objective, is way further from teacher's logit distribution.

**Effect of longer training.** The construction and transfer of richer and more diverse relations mean that VRM may more benefit from longer training. To demonstrate this, we devise a longer training policy (denoted as "LT") than the standard 240-epoch policy in existing KD methods. For our LT policy, the model is trained for 360 epochs and the LR decays by a factor of 10 at the 150th, 180th, 210th, and 270th epochs. All other configurations are kept the same. Tab. 13, VRM indeed benefits from longer training as a 0.21% Top-1 accuracy gain is obtained with LT. In comparison, the performance of other methods plateaued with more training epochs, which is due to overfitting to the training set and a lack of richer guidance signals from the teacher.

**VRM on features.** In this work, we have chosen to construct our virtual relation graphs $\mathcal{G}^{IS}$ and $\mathcal{G}^{IC}$ from network prediction logits $\{\mathbf{z_i}\}_{i=1}^{B}$. In this section, we conduct additional experiments to investigate to what extent VRM can work with features. To this end, we simply reconstruct our graphs from the feature maps $\{\mathbf{f}_i\}_{i=1}^{B}$ right before the final linear layer (denoted as "pooled_feats" in Tab. 14) and in the `mdistiller` codebase. Our virtual relation graphs now become $\mathcal{G}^{IS} \in \mathbb{R}^{B \times B \times D}$ and $\mathcal{G}^{IC} \in \mathbb{R}^{D \times D \times B}$ where D is the dimension of the feature vector. Note that since we no longer work with probability distributions, we remove the Softmax operations that convert predictions to probabilities. Other operations remain unchanged. In Tab. 14, we compare the results of VRM trained using graphs constructed from feature maps with existing methods that

also build relations from the same features (*i.e.*, "pooled_feats"), namely RKD (Park et al., 2019), PKT (Passalis & Tefas, 2018), CRD (Tian et al., 2020), and ReviewKD (Chen et al., 2021b). It can be observed that the performance of VRM deteriorates when applied to features. The reason may be that predicted logits are more compact condensation of categorical knowledge, which is therefore more beneficial for our downstream task. This is particularly so given that VRM does not contain an IM objective that directly matches the logits. As such, VRM works best when applied to logits. Nonetheless, when applied to features, VRM still substantially outperforms all other methods that also work on the very same feature maps. This shows that VRM still encodes better and richer knowledge for distillation compared to the weaker relations transferred by RKD and PKT.

**Comparison to other methods.** We highlight the difference between our method and two KD methods that utilise self-supervised learning, namely SSKD (Xu et al., 2020) and HSAKD (Yang et al., 2021a).

SSKD utilises self-supervision signals via image transformations and pretext tasks for KD. The proposed VRM fundamentally differs from SSKD in at least the following aspects:

① *Motivation*: While both methods involve the utilisation of transformations to produce augmented views of input images, SSKD is directly inspired by and leverage the *pretext task* in self-supervised learning (Chen et al., 2020b). In contrast, our method is pretext-task-free. Instead, VRM is motivated by *transformation invariance regularisation* which was originally popularised in semi-supervised learning (Bachman et al., 2014; Laine & Aila, 2017) and domain adaptation (Berthelot et al., 2021).

② *Training formulation*: A direct consequence of ① is that SSKD's teacher first needs to be re-trained with additional augmentations (which also causes SSKD to use teachers of higher accuracy than ours), followed by a separate fine-tuning stage for the pretext task. These lead to significantly more procedures and computations, whereas VRM is entirely free of such palaver.

③ *Nature of matching objectives*: SSKD is essentially a *hybrid* method that employs both relation matching and instance-to-instance matching objectives, whereas VRM is *purely relation-based* method. In other words, SSKD relies on IM to achieve competitive performance, while VRM involves purely relation-based objectives.

④ *Design choices*: The designs of both methods are vastly different, including but not limited to the formulation of relations and the choices of augmentation policies, relation distance metrics, and model outputs used for computing relations.

HSAKD is another method that makes use of self-supervised learning and transformed views of input images. This method is also fundamentally different from VRM from the following aspects:

① *Motivation*: Like SSKD, HSAKD is also directly motivated by the use of pretext task in self-supervised learning. HSAKD employs rotation prediction as its pretext task. The proposed VRM is free of pretext task learning.

② *Training formulation*: To enable pretext task learning, HSAKD appends auxiliary classifiers to the intermediate features at each stage to perform transformation classification. This means that, akin to SSKD, HSAKD also needs to re-train the teacher model with modified architecture over the pretext task. The auxiliary classifiers also introduce extra parameters. In contrast, the proposed VRM does not involve these additional procedural, parameter, and computational costs.

③ *Nature of matching objectives*: By matching the predictions made by a set of auxiliary classifiers between teacher and student for each sample (as well as matching the final predicted probability distributions between teacher and student), HSAKD is fundamentally a instance matching approach, whereas VRM transfers purely relational knowledge. Moreover, HSAKD employs symmetric matching, which means the matching between teacher and student auxiliary predictions are for the same view of the input samples. By contrast, VRM exploits the relations across asymmetric real and virtual views with different difficulties (*i.e.*, cross-strength matching).

④ *Design choices*: HSAKD also differs from the proposed VRM in terms of specific designs made. For example, HSAKD adopts rotation to construct its pretext task, whereas VRM utilises RandAugment policies. HSAKD also relies on the use of the instance-wise logit matching loss from vanilla KD (Hinton et al., 2015) to reach competitive performance, whereas VRM does not use any instance-matching KD objective and still achieves much more superior performance.

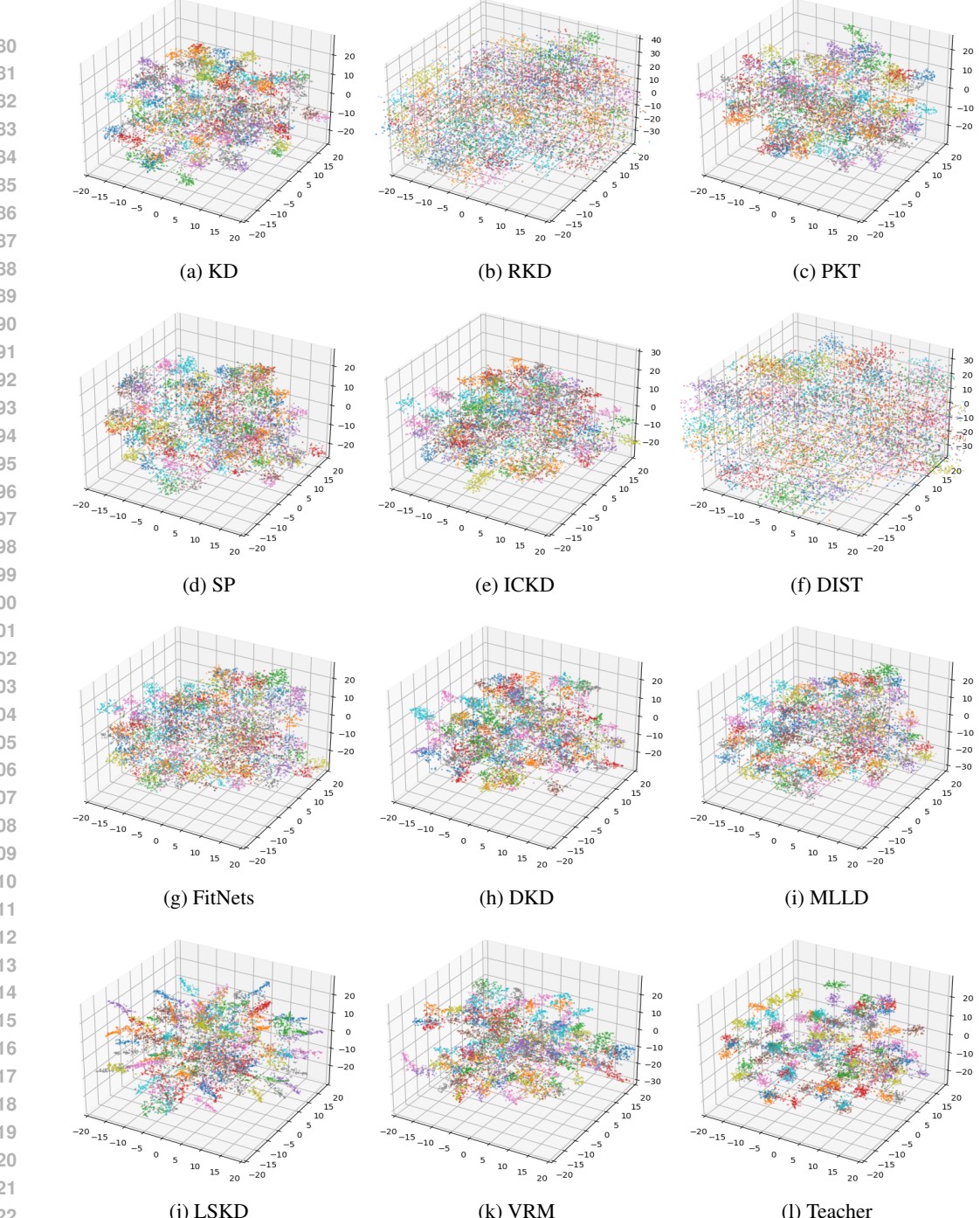

(a) KD       (b) RKD       (c) PKT

(d) SP       (e) ICKD       (f) DIST

(g) FitNets       (h) DKD       (i) MLLD

(j) LSKD       (k) VRM       (l) Teacher

Figure 9: More t-SNE visualisations of features learnt by different KD methods.

## A.7 MORE VISUALISATIONS

**More t-SNE visualisations.** In Fig. 9, we showcase the t-SNE visualisations of embeddings learnt by more KD methods as well as the teacher used (ResNet32×4→ResNet8×4 on CIFAR-100).

**More loss landscape visualisations.** Fig. 10 provides more visualisations of loss landscape for different KD methods.

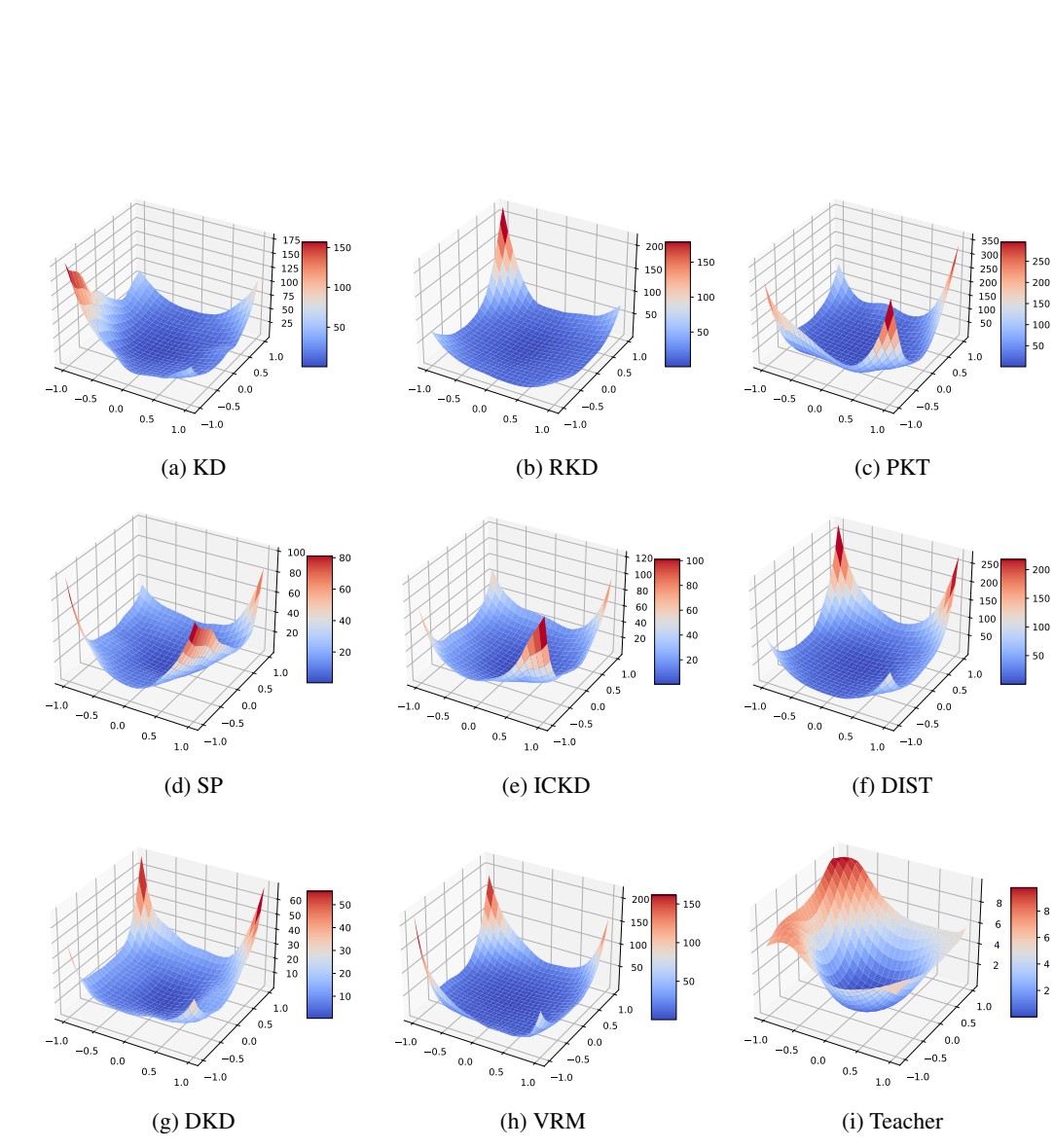

Figure 10: More loss landscape visualisations of different KD methods.

