# OpenReview forum: "VRM: Knowledge Distillation via Virtual Relation Matching"
_ICLR.cc/2025/Conference — ICLR 2025 Conference Withdrawn Submission_

### Official Review · Reviewer_T3S5 · 2024-10-31

**Soundness:** 3
**Presentation:** 4
**Contribution:** 2
**Rating:** 5
**Confidence:** 5

**Summary:**

This work proposes a new relation-based KD method called Virtual Relation Matching (VRM). It is introduced to overcome the problems of relation-based methods, namely the overfitting problem and the  adverse gradients problem. In particular, different from previous works, which utilizes real samples for relation graph construction, VRM adopts virtual samples to build a richer graph. Along with several techniques including distance metric, graph pruning and ZSNorm, VRM outperforms many feature-based KD methods with a significant margin on several popular datasets.

**Strengths:**

This work proposes a new relation-based KD method, VRM. Its strengths are listed as below:
(1) It combines important techniques together and provides a practical framework for relation-based KD.
(2) Extensive experiments are conducted to verify the effectiveness of VRM. VRM shows the best performance under most configurations.
(3) The paper is easy to follow and the idea is clearly stated.

**Weaknesses:**

The weaknesses of the work is listed as below:
(1) There are many techniques and details in VRM framework. It makes VRM seem complex and hard to reproduce. For instance, it has many hyper-parameters to tune.
(2) As the main contribution of the work is "Virtual Relation Matching", we supposed virtual relation to be an important component for final performance gain. However, according to the ablation study, other techniques including ZSNorm and L2Norm contribute significantly. It weakens the contribution of this work.
(3) Similar to (2), graph pruning shows marginal performance gain, though it is claimed as a main contribution of this work.
(4) Important citation missed: Knowledge Distillation via Instance Relationship Graph.

**Questions:**

1. This work utilizes VRM in the logits space. The logits is produced only for the final feature layer or also for several intermediate layers?
2. In section 3.2, it says that "IM matching implies the optimality of relation matching.". Does it imply that IM methods is more potential than RM methods?

---

### Official Review · Reviewer_DFsW · 2024-11-02

**Soundness:** 2
**Presentation:** 3
**Contribution:** 2
**Rating:** 5
**Confidence:** 4

**Summary:**

This study revisits relation distillation and finds the two underlying bottlenecks: prone to over-fitting and the side effect of one spurious sample on others. To address these two drawbacks of relation distillation, the authors propose a virtual relation matching network (VRM). VRM augments the original relation matching by a transformed version of each image in the batch and thus constructs a new relation network of size 2Bx2B for sample matching and 2Cx2C for class matching. A pruning pipeline including redundancy pruning and joint entropy pruning is also adopted to reduce the number of edges and thus significantly reduce memory footprint. Additional logit adapters and logit normalization techniques are also utilized to further improve the efficiency of knowledge transferring process. Experiments on CIFAR-100 and ImageNet are carried out across various architectures and the current method achieves state-of-the-art results on some of them.

**Strengths:**

1. The authors propose two reasons to explain the unsatisfactory performance of previous relation distillation methods, which then leads to the development of the view-augmented version to address the two drawbacks. The motivation is solid and convincing.
2. The current method greatly advances the performance of relation-matching-based distillation methods and achieves on par with or even surpasses some of the recent instance-matching methods. The potential of VRM on heterogeneous distillation is more promising for its clean gap compared to other instance-matching counterparts.
3. The ablation and analysis are presented comprehensively. Tab.5-7 presents the step-by-step progress of this study's performance, including the training dynamics and the analysis of the loss landscape, which contribute to a better understanding of the current methods.
4. The additional analysis of vertex matching and the role of transformation operation help address some of my concerns in advance and are appreciated.

**Weaknesses:**

1. While this study has the potential to bring the relation-matching distillation back into attention, this method sacrifices the elegance of simplicity and instead introduces numerous new modules and hyperparameters, including the balancing loss weight of each module, the pruning criterion and the number of augmentation views. The additional hyperparameters would impair the simplicity and hinder its application in many other tasks.
2. Although the authors state that the current method is not a simple view-augmented relation distillation in the section on further analysis, the proof does not seem convincing enough. As the number of nodes is doubled in VRM, which brings in 1x more additional computational cost, the question would arise if the weakly augmented relation matching network would achieve the same performance given 1x more training budget.
3. The presentation of the two drawbacks of the previous relation-matching network is still intuitive, and the attempted mathematical proof is less vigorously formulated. For example, in the analysis of the effect of the spurious sample on the gradient of other samples, both the input x and guidance y are treated as random variables which leads to 0 correlation between x and y, which clearly does not hold in reality.

**Questions:**

1. Is the method directly applied to the loss function of object detection in Tab 3? Or is it realized by transferring the trained backbone to Imagenet?
2. I would like to hear about the comparison between the current method and an elongated training of view-augmented relation network.

---

### Official Review · Reviewer_k1tV · 2024-11-03

**Soundness:** 4
**Presentation:** 3
**Contribution:** 3
**Rating:** 5
**Confidence:** 4

**Summary:**

This paper introduces Virtual Relation Matching (VRM), a knowledge distillation (KD) approach aimed at overcoming issues in relation-based KD, such as overfitting and sensitivity to spurious responses. VRM constructs affinity graphs that capture inter-sample, inter-class, and inter-view relationships through virtual views, providing a richer set of relational data to guide student model training. The method includes an edge pruning strategy to reduce computational load and enhance robustness. Experimental results demonstrate VRM’s strong performance across CIFAR-100, ImageNet, and MS-COCO datasets, with VRM surpassing both instance- and relation-based KD methods, particularly on heterogeneous teacher-student model pairs and complex object detection tasks.

**Strengths:**

1. The integration of virtual views to enrich relational information represents a fresh perspective on enhancing relation-based KD.
2. The extensive experiments across various datasets and architectures, along with ablation studies, strongly support the validity of VRM’s design choices.
3.The edge pruning mechanism effectively mitigates the impact of spurious relations, allowing VRM to generalize well to test sets.
4. VRM performs well on heterogeneous teacher-student pairs, indicating versatility across tasks and architecture types, with potential benefits in resource-limited scenarios.

**Weaknesses:**

1. Previous GNN-based KD methods, such as [1][2], have shown strong results on various tasks. Without comparisons in the related work or experimental sections, it is difficult to fully assess VRM's effectiveness.
2. Currently, each prediction is assigned only one virtual view. Further analysis of how the number of virtual views affects KD performance and computational cost would provide a more complete evaluation of VRM’s design.
3. Although VRM includes a strategy for pruning unreliable edges, there is a lack of visualization or analysis of the pruned samples to confirm that these edges were indeed unreliable. Comparing VRM’s pruning approach with alternative methods would also clarify its importance in the framework.
4. VRM has been evaluated solely on image tasks. Further exploration into how VRM might be adapted to non-image domains would enhance its potential applicability.

[1]Zhou, Sheng, et al. "Distilling holistic knowledge with graph neural networks." ICCV. 2021.

[2]Zhang, Chunhai, et al. “Multi-scale distillation from multiple graph neural networks.” AAAI. 2022.

**Questions:**

1. Could the authors explore how varying the number of virtual views affects KD performance and computational cost? For example, would adding more views lead to higher accuracy, and how does this impact resource use?
2. Regarding the pruning strategy, could the authors provide some visualization of pruned edges to verify that these samples negatively impact training?
3. How does VRM perform compared to existing GNN-based KD methods? How does the proposed pruning method perform compared to alternative methods? Including these comparisons would highlight VRM's strengths and potential application scenarios.
4. For non-image tasks, would VRM require adjustments to handle other data types? Are there experiments to verify VRM’s potential beyond image-based domains?

---

### Official Review · Reviewer_5Bhp · 2024-11-03

**Soundness:** 2
**Presentation:** 2
**Contribution:** 2
**Rating:** 5
**Confidence:** 4

**Summary:**

1. This paper introduces some improvements to the relation matching (RM) based knowledge distillation methods for image recognition models.
2. A pilot study is included to show why instance-matching methods are often better than relation-matching methods, which is good.
2. The technique contributions include some engineering improvements on the inter-sample and inter-class matching, together with the introduction of using augmented images to facilitate knowledge distillation.
3. The proposed method is tested on several benchmarks, including Cifar-100 and ImageNet image classification, and COCO object detection. The method's effectiveness was validated on some small-sized models, e.g. ResNet-18.
4. Qualitative experiments are included to help authors understand the proposed methods.

**Strengths:**

1. The section on pilot study is interesting and insightful. I hope the authors add more large-scale experiments (e.g., ImageNet experiments) to further validate the claims in this section.
2. The proposed ideas are technically sound, although they are simple and straightforward.
3. The further analysis section (Section 4.4) is interesting to read.

**Weaknesses:**

1. My biggest concern regarding this paper is whether it delivers a meaningful contribution to the community in 2024 given its current experimental setup. Specifically:
- The proposed methods have only been validated on weak baselines, whose training configurations (described in Appendix A.3) are from several yeas ago. In recent years, lots of advancements have been made in developing better training methodologies for training high-performing image classification models, including improved optimizers like AdamW and data augmentation techniques like RandAug. It is well-known that improving weak baselines is relatively straightforward, while achieving improvement on strong baselines is more challenging. The paper does not convincingly show if the proposed method can surpass simple knowledge distillation (KD) techniques, such as basic logit or feature matching, under these modern training settings.
- The architectural choices used in the experiments are outdated. To better demonstrate the effectiveness and generalizability of the proposed method, it is recommended that the authors incorporate experiments with more contemporary baselines, such as ConvNext and Swin Transformer. Even for mobile-optimized models, newer architectures like MobileViT should be evaluated.
- While the authors may claim that they simply follow the experiment settings of some recently published previous works, to me, those works (including this paper) are more like "playing toy KD games on a small scale while completely overlooking the recent advances in a broader scope. Also, it should be noted that there are indeed KD works with up-to-date experiment settings, e.g., [a].
- It is essential to emphasize that this critique does not undermine the potential significance of the work. I acknowledge that knowledge-distillation remains crucial, particularly for deploying models in resource-limited environments. However, to convincingly position this research as a contribution in 2024, significant updates to the experimental settings are necessary.

2. It is unclear whether the reported improvements are primarily from the increased training data introduced by the additional views. A valid question arises as to whether similar performance gains could be achieved by simply incorporating these augmented views during baseline training or while using conventional instance-matching methods?

3. Some proposed techniques yield only minor gains over the baseline. For instance, the primary enhancement in the inter-sample matching component is from employing a new distance metric, which results in only marginal performance improvement (as shown in Table 6).

4. While the paper claims that relational knowledge is particularly valuable for specific architectures, modalities, and tasks (line 48), the effectiveness of the proposed method in these contexts has not been demonstrated. On the contrary, Section 3.2 convincingly illustrates the drawbacks of relation-based methods, raising doubts about the necessity of developing more relation-matching approaches.

[a] Beyer, Lucas, et al. "Knowledge distillation: A good teacher is patient and consistent." Proceedings of the IEEE/CVF conference on computer vision and pattern recognition. 2022.

**Questions:**

All questions are asked in the weakness section.

---

### Note · Authors · 2024-11-14

I have read and agree with the venue's withdrawal policy on behalf of myself and my co-authors.